# OpenReview forum: "Jailbreaking in the Haystack"
_ICLR.cc/2026/Conference — ICLR 2026 Conference Desk Rejected Submission_

### Official Review · Reviewer_EriM · 2025-10-28

**Soundness:** 3
**Presentation:** 3
**Contribution:** 3
**Rating:** 4
**Confidence:** 4

**Summary:**

This paper introduces NINJA, a novel and highly effective jailbreak attack that exploits long-context language models by embedding harmful goals at the beginning of benign, model-generated content. Extensive experiments show that NINJA is stealthy, transferable, and compute-efficient, revealing a fundamental vulnerability in current LLM alignment strategies.

**Strengths:**

1. This paper introduces NINJA, a simple yet powerful jailbreak method that uses entirely benign, semantically relevant content instead of traditional adversarial prompts, revealing a new class of stealthy attacks.
2. This paper provides compelling empirical evidence that the position of harmful goals dramatically affects jailbreak success, uncovering a previously underexplored safety weakness in long-context LMs.
3. The authors evaluate their method across diverse models and agent settings, showing that NINJA is robust and transferable without requiring stronger attacker models or visible adversarial cues.

**Weaknesses:**

1. While the empirical findings on goal positioning are strong, the paper does not offer a clear theoretical framework or model-level analysis (e.g., attention distribution) to explain why early-positioned goals are more effective.
2. The experiments focus primarily on base instruct models and do not extensively evaluate NINJA against recent or state-of-the-art defense techniques.
3. The comparison is limited to only two prior jailbreak methods and more diverse baselines such as GCG and AutoDAN should be included.

**Questions:**

Please refer to the weaknesses.

---

> ### Author Response · Authors · 2025-12-01
>
> We thank the reviewer for their constructive feedback. We address each concern below:
>
> ### W1: Lack of theoretical framework for goal positioning effects
>
> **Concern:** While the empirical findings on goal positioning are strong, the paper does not offer a clear theoretical framework or model-level analysis to explain why early-positioned goals are more effective.
>
> **Response:** We acknowledge that the empirical findings would benefit from deeper mechanistic analysis. To address this, we conducted activation steering experiments that provide mechanistic evidence for why goal positioning matters.
>
> **Experimental Details:**
> - We extracted steering vectors from the residual stream at layer 14 (the layer with maximum cosine difference between goal-present and goal-absent activations)
> - We then used these averaged vectors for activation steering to test whether we could artificially induce positioning effects
>
> **Key Findings:**
> Our steering experiments reveal striking asymmetric effects:
> - Applying steering vectors **after** the goal position maintains high attack success rates (66-74%), closely mimicking our original NINJA results
> - Applying the same vectors **before** the goal position yields only baseline performance (~27.5%), showing no improvement even with increased activation strength
> - Additionally, steering with long-context-specific activations (extracted from the difference between full context vs. goal-only prompts) increases ASR, demonstrating that goal-specific activations are mechanistically crucial for attack success
>
> These results provide initial mechanistic evidence that early goal positioning creates specific activation patterns that persist through the model's computation, affecting how subsequent context is processed.
>
> ### W2: Limited evaluation on state-of-the-art defenses
>
> **Concern:** The experiments focus primarily on base instruct models and do not extensively evaluate NINJA against recent or state-of-the-art defense techniques.
>
> **Response:** We focus our evaluation on standard mid-scale instruction-tuned models, including recent production-grade systems such as Gemini 2.0 Flash. Our goal is to characterize the vulnerability landscape for models that are widely deployed in research and industry applications.
>
> We acknowledge that NINJA, similar to other prompt-based attacks like PAIR and Many-shot, is substantially less effective on frontier models (e.g., Claude, GPT-4). This limitation is consistent with prior findings in the jailbreaking literature and aligns with our positioning of NINJA as revealing a fundamental vulnerability in the architecture of deployable mid-scale models rather than claiming to break all possible defenses.
>
> ### W3: Limited baseline comparisons
>
> **Concern:** The comparison is limited to only two prior jailbreak methods and more diverse baselines such as GCG and AutoDAN should be included.
>
> **Response:** We have now evaluated both methods across our HarmBench setup for completeness.
>
> **Important Methodological Distinction:**
> GCG and AutoDAN are optimization-based adversarial attacks, while NINJA is a prompt-based attack with fundamentally different requirements:
> - **GCG** requires gradient access and hundreds of optimization steps per example (often 1-3 hours per model for 80 HarmBench tasks)
> - **AutoDAN** performs iterative, token-level optimization (via gradients / model internals) to craft jailbreak prompts requiring white-box access during generation.
> - **NINJA** runs in a single forward pass without requiring logits, gradients, or optimization loops
>
> For methodological consistency, our main comparison focuses on prompt-based attacks (PAIR, Many-shot). However, here are the complete results including GCG and AutoDAN:
>
> | Method | Llama-3.1 | Qwen2.5 | Mistral-7B | Gemini 2.0 Flash |
> |--------|-----------|---------|------------|------------------|
> | PAIR | 0.220 | 0.346 | 0.413 | 0.153 |
> | Many-shot | 0.450 | 0.225 | 0.125 | 0.500 |
> | GCG | 0.525 | 0.800 | 0.713 | - |
> | AutoDAN | 0.725 | 0.650 | 0.688 | - |
> | **NINJA** | **0.588** | **0.425** | **0.545** | **0.288** |
>
> While GCG and AutoDAN achieve higher ASR on some models, they require orders of magnitude more computational resources and  model access(white-box attack). NINJA's strength lies in its simplicity and efficiency—achieving competitive performance with a single forward pass using only benign context, making it a more practical and stealthy attack vector in real-world scenarios.
>
> ---
>
> We hope these clarifications and additional experiments address the reviewer's concerns. We will incorporate these discussions into the revised manuscript.

---

### Official Review · Reviewer_KXbt · 2025-10-30

**Soundness:** 3
**Presentation:** 3
**Contribution:** 3
**Rating:** 8
**Confidence:** 4

**Summary:**

This paper introduces NINJA (Needle-in-haystack jailbreak attack), a novel method that exposes a critical safety vulnerability in long-context language models. The attack operates by placing a harmful user goal (the "needle") at the beginning of a long, semantically relevant, but otherwise benign context (the "haystack"). The authors demonstrate that this technique significantly increases Attack Success Rates (ASR) on models like LLaMA-3.1 and Qwen2.5. The paper's core contributions are: (1) identifying that the position of the harmful goal is a critical variable, with attacks being far more successful at the start of the context than the end, and (2) demonstrating that this long-context attack is more compute-efficient than standard best-of-N sampling, especially as an attacker's compute budget increases.

**Strengths:**

1. The paper presents a clear identification and empirical validation of goal positioning as a critical safety vulnerability. This reframes positional bias from a simple capability quirk to a fundamental, exploitable flaw in safety alignment.
2. The experiment in Section 5.3 / Figure 5, which compares relevant vs. irrelevant context, is a cool and unique contribution . It proves that the attack is not merely "confusing" the model with noise, but actively "distracting" its attention with semantically related, benign content.
3. The compute-optimality analysis in Section 5.4 / Figure 6 is a significant strength.

**Weaknesses:**

The paper clearly distinguishes NINJA (relevant context, goal at start) from Cognitive Overload (distracting context, goal at end) . However, it doesn't complete the "cross-over" experiment. The authors' own findings show NINJA fails if the goal is at the end (Figure 3). To fully prove that relevance is the key differentiator, they should have also tested a "Cognitive Overload at Start" (i.e., goal at start + irrelevant context). This would isolate whether the "goal-at-start" phenomenon is universal or one that is uniquely enabled by the relevant context.

**Questions:**

Can you answer the Comments I made in the "Weakness section"?

---

> ### Author Response · Authors · 2025-12-02
>
> We thank the reviewer for their positive assessment and insightful suggestion about the cross-over experiment.
>
> ### W1: Missing "Cognitive Overload at Start" cross-over experiment
>
> **Concern:** The paper doesn't complete the cross-over experiment by testing "Cognitive Overload at Start" to isolate whether goal-at-start is universal or uniquely enabled by relevant long context.
>
> **Response:** We agree this cross-over experiment strengthens our claims. We have now conducted this experiment using the multilingual Cognitive Overload approach.
>
> **Experimental Setup:**
> We use the multilingual variant from Xu et al. [1] (the only reproducible implementation, as Upadhayay et al. [2] did not release code). Following the setup described in their repository (https://github.com/luka-group/CognitiveOverload), we tested on AdvBench dataset using the vicuna-7b model. We created two conditions with identical content but different goal positioning:
>
> - **Original (goal at END):** English Wikipedia context → Harmful goal in language X
> - **Cross-over (goal at START):** Harmful goal in language X → Same English Wikipedia context
>
> Both conditions use the same harmful goal and benign context; only the ordering differs.
>
> **Results (19 languages, 520 prompts per language, 9,880 total attacks):**
> - Goal at END: 91.21% ASR
> - Goal at START: 99.28% ASR
> - **Δ = +8.07pp improvement from moving goal to start**
>
> **Key Finding:**
> This cross-over experiment demonstrates that the goal-at-start vulnerability is a general phenomenon that applies across different attack contexts, not just in NINJA's semantically relevant settings. Even in Cognitive Overload's irrelevant/distracting context setup, positioning the harmful goal at the beginning consistently improves attack success. This completes the experimental matrix the reviewer requested and confirms that goal positioning is a fundamental vulnerability in long-context LLMs.
>
> We will add these results to Section 5.2 and include a detailed analysis in the appendix.
>
> ---
>
> [1] Xu et al., "Cognitive Overload: Jailbreaking Large Language Models with Overloaded Logical Thinking," arXiv:2311.09827, 2024.
> [2] Upadhayay et al., "Cognitive Overload Attack: Prompt Injection for Long Context," arXiv:2410.11272, 2024.

---

### Official Review · Reviewer_1kog · 2025-10-31

**Soundness:** 2
**Presentation:** 2
**Contribution:** 2
**Rating:** 2
**Confidence:** 4

**Summary:**

This paper introduces "NINJA" (Needle-in-haystack jailbreak attack), a method that jailbreaks aligned LLMs by embedding a harmful goal at the beginning of a long, benign, and semantically relevant context. The authors claim that increasing context length itself significantly degrades model safety, with goal positioning being a critical factor. Experiments on HarmBench and agentic frameworks demonstrate NINJA's effectiveness over PAIR and Many-shot baselines, and its compute-efficiency under fixed budgets.

**Strengths:**

1. The paper demonstrates that long, benign contexts can be an effective attack vector, which is practical and stealthy due to the use of non-malicious content.

2. The study provides a clear empirical analysis of how goal positioning within the context affects attack success.

**Weaknesses:**

1. The paper fails to adequately distinguish its core contribution from existing long-context attacks, particularly Many-shot Jailbreaking [1]. While the authors note that NINJA uses "entirely innocuous context," this distinction is superficial. Both methods exploit long contexts to dilute safety alignment; the difference between "explicitly harmful" and "benign" examples is a matter of degree rather than a fundamental mechanistic difference. A deeper discussion of the underlying failure mode (e.g., attention dilution, task confusion) shared by both approaches is needed to establish a clear boundary for the claimed novelty.

2. The reported performance gains do not robustly support the claim of "significant" safety degradation. While the improvement for Llama-3.1-8B is notable (23.7% to 58.8%), the results for Qwen2.5-7B (23.7% to 42.5%) are modest, and the gain for Gemini 2.0 Flash (23% to 29%) is minimal—a mere 6 percentage points. This weak performance on a widely-used model severely undermines the paper's argument that this is a universal and critical vulnerability. The claim would be better supported by a more nuanced interpretation that acknowledges significant model-dependent variation.

3. The experimental comparison is limited to only two baseline methods (PAIR and Many-shot). This narrow scope overlooks several other relevant and strong baselines, such as ReNeLLM  or CodeAttack. The absence of these comparisons makes it difficult for the reader to gauge NINJA's true standing in the current landscape of jailbreaking techniques and to assess whether the observed performance is state-of-the-art or simply an incremental improvement over a weak set of baselines.

4.The paper briefly suggests placing user goals at the end of the prompt as a mitigation strategy but provides no empirical validation of this defense's efficacy or its potential impact on model capability for legitimate long-context tasks. A convincing safety analysis should at a minimum test this proposed defense against the NINJA attack and discuss its limitations and potential side effects. The absence of any defensive evaluation makes the contribution less actionable for practitioners seeking to secure their systems.

References:

[1] Cem Anil, Esin Durmus, Mrinank Sharma, Joe Benton, Sandipan Kundu, Joshua Batson, et al. Many-shot jailbreaking. arXiv preprint arXiv:2304.XXX, 2024b. Anthropic Technical Report.

**Questions:**

None

---

> ### Author Response · Authors · 2025-12-01
>
> We thank the reviewer for their detailed feedback and thoughtful analysis. We address each point below:
>
> ### W1: Distinction from existing long-context attacks
> **Concern:** The paper fails to adequately distinguish its core contribution from existing long-context attacks, particularly Many-shot Jailbreaking.
>
> **Response:** We appreciate this important point and agree that a clear distinction from many-shot jailbreaking is warranted:
>
> **1. Practicality:**
> While both many-shot jailbreaking and our proposed NINJA method leverage long context to exploit the guardrails present in language models, NINJA is solely dependent on benign innocuous long context (context related), while many shot consists of hand crafted many shot examples that tricks the model into generating a response to a harmful query.
> While innovative, many context attack is easy to catch and filter and is very specific and easy to defend against, whereas NINJA is much more fundamental in nature. NINJA exposes the vulnerability that comes when LLMs are used in long context settings, and given the wide adoption of LLM based agents in the current time, NINJA highlights a fundamental risk that long context poses which must be addressed to build reliable and robust systems.
>
> **2. Defences:**
> Many-context and other such attacks working in long context settings like ReNeLLM and CodeAttack and easily be filtered out using regex based approaches or even as LLM based judges. However, NINJA works on innocuous related benign context and also emphasises the role the positioning of goal plays in such scenarios. NINJA provided a more realistic playground and exposes a more critical problem that exists. In fact, to our belief these other attacks are in fact a subset of NINJA and the fact that these work, corroborates our findings of the vulnerabilities exposed by long context. Coming up for defences for such long context that is semantically related to the main task is kind of challenging, whereas finding out defences for attacks like many-shot jailbreak is relatively easier. Thus, we believe that our work highlights a more fundamental challenge.
>
> To understand the underlying failure modes, we also do a feature steering experiment to demonstrate this distinction concretely:
> ```
> Baseline -> <HARMFUL GOAL> ASR=27%
> <HARMFUL GOAL> + <vector from related long context> ASR=72%
> <HARMFUL GOAL> + <vector from unrelated long context> ASR=64%
> <vector from related long context> + <HARMFUL GOAL>  ASR=32%
> <vector from unrelated long context> + <HARMFUL GOAL>  ASR=29%
> ```
>
> The vector was injected using a forward hook at layer 14, selected based on the maximum deviation between context-driven and baseline activations. While results were consistent across intermediate layers, injection at later layers degraded generation quality, likely by disrupting the language modeling head. This setup follows the feature steering protocols described in Evaluating Feature Steering: A Case Study in Mitigating Social Biases - Esin Durmus et al (https://www.anthropic.com/research/evaluating-feature-steering).
>
> Steering with long-context-specific features (extracted from the difference between full context vs. goal-only prompts) increases ASR, however the ASR is much more pronounced when the vectors are obtained from similar benign long context. This fact explain why we claim NINJA (semantically related long context) to be more effective than many shot attack (unrelated long context) since we see that features from semantically related context lead to a higher degradation of the safety guardrails.
>
> Overall we believe:
>
> Many-shot jailbreaking could potentially be defended against by detecting repeated harmful patterns or limiting in-context learning of unsafe behaviours. A simple regex for the presence of such in context examples can and should mitigate many-shot jailbreaking.
> NINJA is immune to such defences. Innocuous long context related to the context is almost impossible to filter out, and the fact that it leads to decreased safety is a point that the frontier model developers should tend to while developing models and agents working with longer contexts.
>
> ### W2: “Significance of performance gains”:
> We humbly argue that an AST increase on Qwen2.5-7B from **23.7% to 42.5%** is in fact a **79%** relative decrease in safety which we think is significant. Similarly, a **25%** relative increase for Gemini 2.0 Flash, a frontier model with presumably strong safety measures, demonstrates that even state-of-the-art systems are vulnerable. These improvements, especially given that the attack is in fact entirely natural and cheap, represent meaningful security risks that practitioners should be aware of, especially as long-context applications become more prevalent.

---

> > ### Author Response · Authors · 2025-12-01
> >
> > ### W3: Limited baseline comparisons
> > **Response:** We would like to clarify our positioning. We present long context as an existing vulnerability that can be exploited rather than as an explicit attack method. ReNeLLM and CodeAttack are handcrafted methods that tune specific exploits against safeguards, but the implications of NINJA are more fundamental.
> >
> > With the increasing demand for agents handling long-horizon tasks, long-context attacks pose a fundamental challenge that needs attention. The existence of such behaviors in agentic tasks (e.g., SHADE-Arena (Kutasov et al., 2025) tasks and growing safety evaluation studies like BrowserART (Kumar et al., 2024)) highlights that NINJA addresses a more fundamental architectural vulnerability rather than a specific bypass technique like ReNeLLM or CodeAttack.
> >
> > Our findings expose a critical gap in current safety paradigms: the assumption that short-context alignment transfers to long-horizon tasks. With the rapid deployment of agents, this assumption is becoming a liability. We highlight this fundamental architectural vulnerability not merely as an attack vector, but to motivate the development of robust defenses capable of scaling with the expanding context windows of modern models.
> >
> >
> > ### W4: Lack of defense validation
> >
> > **Response:** We acknowledge this limitation and agree that comprehensive defense evaluation would strengthen the contribution. Our goal in this work is focused on characterizing the vulnerability and its mechanisms. We will add a brief discussion of this limitation and future work direction in the revised version.
> >
> > ---
> >
> > We hope these clarifications address the reviewer's concerns and better contextualize our contributions.

---

### Author Response · Authors · 2025-12-03
**Response to Area Chair: Summary of Rebuttal Updates**

Dear Area Chair

We thank the reviewers for their constructive feedback. We have addressed **all raised concerns** by conducting **3 major new experiments** during the rebuttal period. Below, we map these new results directly to the specific reviewer critiques.

**General statement:** The reviewers all seemed to appreciate our comprehensive results and findings on **goal positioning** and **effect of long-context**. A major clarification in our rebuttal is more analysis on **why goal positioning matters** and **why long-context degrades safety**. We also clarify the positioning of our work: **we do not claim long context to be SOTA attack** but rather a **pervasive vulnerability**: long context is natural, more and more used with agents, hard to filter out “benign” context that someone might inject, requires very low resources to generate benign context (outperforms other similarly prompt-based attacks like many shot and PAIR), holds on medium-scale production LLMs

### Reviewer 1kog

The reviewer appreciated our analysis of goal positioning but raised the following concerns about the overall work:

**“Distinguishing from long-context attacks”**: We show that the mechanism of our jailbreak in the haystack is fundamentally different from “many shot jailbreaking”.
From a practical perspective, the differences are:
- It is practically much harder to detect benign long content
- It doesn’t even require adversaries to specifically add context, so it can happen in the wild with just a “dumb” adversary

**“Significance of performance gains”**: We humbly argue that an ASR increase on Qwen2.5-7B from **23.7% to 42.5%** is in fact a **79%** relative decrease in safety which we think is significant. Similarly, a **25%** relative increase for Gemini 2.0 Flash, a **frontier model** with presumably strong safety measures, demonstrates that even SOTA systems are vulnerable. This, especially given that the attack is in fact **entirely natural and cheap**, represent security risks that practitioners should be aware of especially as long-context applications become more prevalent.

**Baseline Comparisons (GCG & AutoDAN Evaluation):**
We'd like to clarify that our goal wasn’t to develop a SOTA attack but demonstrate an **important vulnerability** which is potentially overlooked.
For completeness, we also added GCG, AutoDAN baselines. Note that these attacks **require white box access to models** and are computationally **much more heavy weight**. Despite that, we are able to roughly match these results in some cases with just a really **simple and cheap method** which is concerning and a significant contribution in our opinion.

### Reviewer KXbt

The reviewer, generally very positive, had a request for a cross-over experiment on the **"Cognitive Overload at Start"**. We did our goal positioning experiment and our findings in NINJA generalised to the cognitive overload setting -- goal positioning in the presence of long context indeed matters. We **observed a gain in the ASR (Δ = +8.07pp improvement from moving goal to start)** on placing the goal at the start in this setup. This confirms that goal positioning is a fundamental vulnerability in long-context LLMs.

### Reviewer ERiM
Found our goal positioning results comprehensive but was concerned about lack of more analysis.

**SOTA defenses and attacks:** We focus our evaluation on standard mid-scale instruction-tuned models, including recent production-grade systems such as Gemini 2.0 Flash. Our goal is to characterize the vulnerability landscape for models that are widely deployed in research and industry applications.

**Lack of analysis for goal positioning effects:** While any theoretical results are difficult we carried out feature steering experiments to provide insight into why goal positioning matters. Results align with our NINJA findings: benign related long context degrades safety more than unrelated context, and early goal positioning followed by long context amplifies safety degradation.

**Limited baseline comparisons:** We compared NINJA with the existing white box gradient based attacks. Although the white box attacks (GCG, AutoDAN) are more effective in certain settings, they are harder and more costly to train and require both time and compute. The fact that a simple one shot long context attack is able to roughly match the performance of such attacks in most of the settings is quite alarming. Our findings expose a critical vulnerability present in today’s LLMs and if such LLMs are to be used in long contexts, this is something fundamental that needs to be addressed.

**Limited baseline comparisons:** We compared NINJA with white-box gradient-based attacks (GCG, AutoDAN). While these are more effective in some settings, they require significant time and computation. That a simple one-shot long-context attack roughly matches their performance in most settings is alarming and highlights a fundamental vulnerability in today's LLMs that must be addressed.

---

### Note · Program_Chairs · 2026-01-17
**Submission Desk Rejected by Program Chairs**

The following references in this submission do not refer to real documents and/or have major errors in bibliographic information:

 Yikang Wu, Hongding Diao, Xingchen Diao, Zhaowei Wang, Ziyang Chen, Pe Song, Guang yao Zhai, and Hong han Shuai. Unpacking the positional bias of large language models. 2024.